# Clinical Evaluation of the Pancreatic Cancer Microenvironment: Opportunities and Challenges

**DOI:** 10.3390/cancers16040794

**Published:** 2024-02-15

**Authors:** Julianne M. Szczepanski, Mark A. Rudolf, Jiaqi Shi

**Affiliations:** Department of Pathology and Clinical Labs, University of Michigan, Ann Arbor, MI 48109, USA; jmboyle@umich.edu (J.M.S.); mrudolf@med.umich.edu (M.A.R.)

**Keywords:** pancreatic ductal adenocarcinoma, pathology, management, biomarker, tumor microenvironment

## Abstract

**Simple Summary:**

Pancreatic ductal adenocarcinoma (PDAC) continues to be a significant challenge in cancer management, with predictions indicating it will become the second leading cause of cancer-related death by 2030. The persistent poor outcomes in PDAC are partly attributed to a complex tumor microenvironment (TME) that inhibits anti-tumor immunity, restricts the penetration of therapeutics, and facilitates cancer dissemination. Only a few aspects of the TME are routinely documented in the clinical setting, but this may evolve as new TME-directed therapies are developed. In this review, we cover relevant aspects of modern PDAC management, summarize TME-related biomarkers and clinical trials, and discuss the potential challenges associated with evaluating the TME in a clinical context.

**Abstract:**

Advances in our understanding of pancreatic ductal adenocarcinoma (PDAC) and its tumor microenvironment (TME) have the potential to transform treatment for the hundreds of thousands of patients who are diagnosed each year. Whereas the clinical assessment of cancer cell genetics has grown increasingly sophisticated and personalized, current protocols to evaluate the TME have lagged, despite evidence that the TME can be heterogeneous within and between patients. Here, we outline current protocols for PDAC diagnosis and management, review novel biomarkers, and highlight potential opportunities and challenges when evaluating the PDAC TME as we prepare to translate emerging TME-directed therapies to the clinic.

## 1. Introduction

By 2030, pancreatic ductal adenocarcinoma (PDAC) is projected to be the second leading cause of cancer-related death [1]. There remains a significant need to improve therapeutic outcomes for PDAC patients.

As the era of precision oncology unfolds, it is becoming standard for modern pathology laboratories to perform next-generation sequencing (NGS) and other ancillary studies to diagnose, subtype, and identify actionable mutations within tumors. Not only is it crucial to characterize the cancer cells themselves, but also the tumor microenvironment (TME). The TME includes the non-tumor cells, extracellular matrix, and local microbiota [2], as well as the biochemical, biophysical [3], and bioelectric conditions of the stroma and interstitial fluid that exist in and around a tumor [4,5]. Our basic understanding of the PDAC TME is rapidly evolving, and a combination of experimental and observational evidence indicates that the TME affects clinical outcomes in PDAC by influencing its outgrowth and responses to therapy [6,7].

Whereas cancer cell profiling has become sophisticated, current clinical protocols to evaluate the tumor microenvironment lag. The standard methodology involves manual morphologic identification and reporting of perineural invasion (PNI) and lymphovascular invasion (LVI) on routine histologic sections. Various TME cell types are not routinely identified, quantified, phenotyped, or reported in standard pathology practice. Yet, implementation of future TME-targeted therapies may require a more personalized evaluation of each tumor’s specific microenvironmental characteristics.

Here, we review how PDAC specimens are currently evaluated through histology and sequencing. We then briefly discuss actionable genetic alterations in PDAC tumor cells. We next summarize advances in the basic understanding of the PDAC TME, with a focus on current and prospective biomarkers that may impact the clinical laboratory (Figure 1). Our aim is to present an overview of our expanding understanding of the PDAC TME, specifically TME heterogeneity, and to contrast that with the simplistic methods of TME evaluation in current clinical practice. To harness the potential of TME-directed therapies in PDAC, both basic scientists and practicing pathologists may wish to consider whether key TME changes are universal, and, if not, how to evaluate and account for this heterogeneity.

## 2. Overview of PDAC Management and Specimen Acquisition

Patients with PDAC usually present with fatigue, weight loss, abdominal pain, dark urine, or jaundice [33]. Less commonly, patients can present with an incidentally discovered pancreatic mass or biliary abnormality on imaging [34]. After dedicated computed tomography (CT) or magnetic resonance imaging of the pancreas for staging and evaluation of tumor resectability, additional imaging is pursued to evaluate for metastatic disease [35,36,37]. Histologic confirmation of malignancy follows. In the absence of obvious metastases, patients usually undergo endoscopic ultrasonography (EUS) and EUS-guided biopsy [36,37,38]. Fine-needle biopsy is superior to fine-needle aspiration and cytologic evaluation [38]. Conversely, if metastatic (approximately 50% of patients) [39], biopsy from a more easily accessible metastatic site is preferred. For all needle biopsies, it is recommended that two extra needle passes be performed to collect material for future NGS analysis. For patients with biopsy-proven metastatic disease, recommendations include genetic testing for inherited mutations, molecular profiling of tumor tissue, and complete staging, followed by clinical trial placement, systemic therapy, or palliative care as clinically indicated.

Patients without detectable metastases are evaluated for resectability of the primary tumor [35]. Preferably, resections are performed at institutions that perform at least 15–20 annual procedures [40,41,42,43]. Those with resectable disease undergo surgery with or without neoadjuvant therapy and may go on to receive adjuvant treatment. Those with borderline resectable disease may be referred to a high-volume center for evaluation and consideration of endoscopic retrograde cholangiopancreatography with stent placement and neoadjuvant therapy with repeat assessment. Therapies may include clinical trials, a combination of systemic or induction chemotherapy and/or chemoradiation, or stereotactic body radiation therapy, depending on whether the patients are candidates for induction chemotherapy [44,45,46]. Genetic testing for inherited mutations and molecular profiling of tumor tissue may be recommended in these patients as well. Most treatment protocols require biopsy confirmation of adenocarcinoma. However, the lack of a diagnostic biopsy should not delay surgical resection when the clinical suspicion of pancreatic cancer is high.

As outlined above, biopsy material is important for diagnostic purposes, but also to provide tissue for molecular profiling. Currently, it is recommended to specifically test for potentially actionable somatic findings including, but not limited to, fusions (*ALK*, *NRG1*, *NTRK*, *ROS1*, *FGFR2*, and *RET*), mutations (*BRAF*, *BRCA1/2*, *KRAS*, and *PALB2*), amplifications (*HER2*), microsatellite instability (MSI), mismatch repair deficiency (dMMR), and tumor mutational burden via an FDA-approved and/or validated next-generation sequencing (NGS)-based assay [8]. RNA sequencing assays are preferred for detecting fusions.

Although targeted molecular testing is preferably performed on tumor tissue, “liquid biopsy” of circulating tumor DNA (ctDNA) testing is an emerging option for diagnosis, prognosis, disease monitoring, and potentially screening. Given its high prevalence in PDAC, mutant *KRAS* is a promising biomarker for this purpose. A 2017 study found that *KRAS* mutations were detectable in 30% of patients with resectable PDAC, and the rate of detection was proportional to tumor size [47]. Combining the assay with CA 19-9 and three other plasma protein markers further increased the sensitivity to 64%, with specificity exceeding 99%. The strategy was incorporated into an expanded ctDNA assay for eight different cancer types, termed CancerSEEK [48].

## 3. Genomic Drivers of PDAC

Approximately 12–25% of pancreatic cancers contain actionable molecular alterations, and patients who receive a matched therapy have significantly improved overall survival [49]. 

Oncogenic *KRAS* mutations are likely the initial driver mutation in the majority of PDAC cases, regardless of whether tumors arise from PanIN or cystic mucinous neoplasms. Small-molecule inhibitors targeting *KRAS*^G12C^ (sotorasib and adagrasib) are approved in the United States; this mutation occurs in the majority of non-small cell lung carcinomas but only 1–3% of PDAC cases (<3%) [50]. Approaches targeting the more common *KRAS* mutations in PDAC, such as *KRAS^G12D^* (~40%), *KRAS^G12V^* (~40%), and *KRAS^G12R^* (~15–20%), are in development [51]. In particular, there has been recent progress targeting *KRAS^G12D^* with a current Phase 1 clinical trial in progress (NCT05382559) [52,53,54], and the first pan-RAS inhibitor (RMC-6236) is being evaluated in phase I clinical trials with promising preliminary results (NCT05379985). *KRAS* genotyping will be an important aspect of pathologic testing of PDAC as these inhibitors gain approval, especially if they are allele specific. 

The approximately 10% of PDAC tumors that have wild-type *KRAS* have a distinct molecular profile and are more likely to harbor oncogenic fusion events [55], which are frequently actionable. This has led Topham et al. to advocate for *KRAS* mutation panel analysis with subsequent NGS testing for *KRAS* wild-type tumors.

Small percentages of PDAC tumors bear other targetable molecular alterations. For example, the RET inhibitor selpercatinib is approved for cancers harboring a *RET* gene fusion, which was identified in 0.6% (1/160) of PDAC cases in one study [56]. *NTRK* gene fusions are now amenable to targeting with larotrectinib and entrectinib; however, they are also rarely identified in PDAC, occurring in less than 1% of cases [57,58].

Around 5–7% of PDAC patients carry a germline mutation in genes associated with Fanconi anemia (*BRCA1*, *BRCA2*, and *PALB2*), affecting proteins crucial for homologous recombination DNA repair [59]. Tumors with loss of the remaining wild-type allele are more sensitive to platinum-based chemotherapy and PARP inhibitors [60]. A clinical trial in PDAC patients with germline *BRCA1/2* mutations showed improved progression-free survival with PARP inhibitors [61]. In addition to germline mutations, some tumors with somatic mutations of *BRCA1/2* and *PALB2* and established homologous repair deficiency also show benefit from PARP inhibitors [62].

The genes belonging to the chromatin-remodeling and switch/sucrose non-fermentable (SWI/SNF) complex are among the most commonly altered class of genes in PDAC [63,64,65,66], often through deletions [67]. Considering how often PDAC cases involve changes in these genes, a handful of experimental therapeutic options have been proposed [68,69,70]. Tumors with COMPASS-like complex gene alterations exhibit squamous morphology or are poorly differentiated and are associated with poor survival [71]. 

Tumors with dMMR arise sporadically due to inactivating mutations in *MLH1*, *PMS2*, *MSH2*, and *MSH6*, or due to *MLH1* promoter methylation; they also arise in the context of Lynch syndrome. Such tumors are characterized by microsatellite instability (MSI-high). However, only 0.7–2% of PDAC are dMMR [72,73]. dMMR PDAC is associated with medullary or mucinous/colloid histology [74,75,76]. Across a variety of cancers, tumors with dMMR are generally more responsive to immune checkpoint blockade, which is attributed to a high rate of frameshift mutations and thus production of mutation-associated neoantigens [72]. While an initial study including a limited number of dMMR PDAC cases reported that checkpoint inhibitor therapy had a promising overall response rate in five of eight cases [72], a subsequent trial reported a response rate of 18.2% (four of twenty-two) among non-neuroendocrine pancreatic cancer cases with dMMR [77]. It has been shown that dMMR tumors are dependent on the RecQ DNA helicase *WRN* for genome maintenance, and loss of WRN is a synthetic lethality in dMMR cells [78]. This discovery has led to the potential for targeting WRN in patients with dMMR tumors who are refractory to immune checkpoint blockade [79]. A first-in-class WRN inhibitor (RO7589831) is already in a phase I clinical trial against dMMR/MSI-high tumors (NCT06004245).

Other biomarkers that are associated with improved response to immune checkpoint blockade across a variety of tumor types include high tumor mutational burden, positive PD-L1 expression (in cancer cells and cells in the TME), and T-cell-inflamed gene expression profile [28]. In the KEYNOTE-028 trial (NCT02054806), which examined the efficacy of checkpoint blockade with the anti-PD-1 mAb pembrolizumab in various PD-L1 positive tumors, including 24 PDAC cases, PDAC showed the lowest overall response rate across 20 tumor types (zero responders), which precluded evaluation of these biomarkers [28]. Similarly, a study of 65 PDAC cases reported a 3.1% response rate to checkpoint inhibition, the low rate again precluding assessment of biomarkers [80]. Thus, it remains unclear whether additional biomarkers beyond microsatellite instability/dMMR predict responsiveness to checkpoint inhibition in PDAC.

Mutations in the DNA replication and proofreading enzymes *POLE* and *POLD1* lead to “ultramutated” tumors with improved prognosis and, in limited studies to date, an improved response to immune checkpoint inhibition [81,82]. Again, such mutations appear to be rare in PDAC; one study found no pathogenic *POLE* exon 9 or 13 mutations in a cohort of 115 cases [83].

*SMAD4* and *TP53* are both considered late alterations in PDAC development [84]. *TP53* and *SMAD4* alterations may confer prognostic significance and can be assessed via immunohistochemical (IHC) stains [85]. *CDKN2A* alterations, which can be detected by p16 protein loss by IHC, are also very common in PDAC and are associated with aggressive tumors [84]. These tumors are associated with the adenosquamous variant of PDAC, as well as tumors with foci of unconventional morphologic features such as conspicuous cribriform, clear-cell, papillary, gyriform, or micropapillary features [84].

As the list of actionable molecular targets lengthens, pathologists will play a greater role in IHC interpretation, assessment of tissue adequacy for molecular studies, and directing appropriate testing based on the amount of available tumor tissue.

## 4. Transcriptional Subtypes of PDAC

Numerous efforts to molecularly characterize PDAC have led to the recognition of two major subtypes, termed “classical” and “basal-like” [86]. The classical subtype is GATA6-expressing and KRAS-dependent [87]. On histologic evaluation, mucinous tumors more commonly fall into the classical subtype [6]. 

The basal-like subtype, which accounts for approximately one third of PDAC cases, has been shown to lose expression of *GATA6*, and these tumors are less responsive to first-line 5-fluorouracil-based therapies [88,89,90]. *GATA6* RNA in situ hybridization is a surrogate biomarker for basal-like PDAC and may eventually become routine for evaluation of PDAC specimens [91]. These tumors tend to show higher histologic grade compared to the classical subtype [6]. 

Although the classical and basal-like subtypes were initially described as distinct entities, single-cell RNA sequencing studies of PDAC tumors have uncovered that the majority of tumors are composed of a mixture of the two subtypes [7,92]. In one set of experiments, a specific cell clone isolated from a metastatic PDAC lesion transitioned from a basal-like signature to a classical-like signature in ex vivo organoid culture [7]; adjusting media conditions could induce cell clones to transition between transcriptional subtypes and altered responses to chemotherapy [7]. These findings underscore the marked effect of the TME on gene expression and therapeutic response. A summary of genomic drivers and alterations studied in PDAC, and the clinical implications and pathologist’s role for each, is provided in Table 1.

## 5. The PDAC TME

While determining genetic alterations within PDAC tumor cells is crucial, the TME is also recognized to play a major role in PDAC. Several therapies targeting the TME are presently being tested in clinical trials (Table 2). To date, few trials have employed biomarkers to select patients who are most likely to benefit from treatment. We anticipate that TME evaluation will become increasingly important as more is learned about TME heterogeneity between patients, particularly as trials progress into later stages that aim to establish therapeutic efficacy. 

### 5.1. General Considerations for TME Evaluation in PDAC

As outlined above, many treatment decisions in PDAC are based on analysis of biopsy material from primary or metastatic sites. These core biopsies also provide the first opportunity to directly assess the TME, raising important questions and possible challenges (Figure 2).

#### 5.1.1. Sampling the TME in Small Biopsies

Do biopsies sample enough material to assess the TME? Tumor cellularity varies among cancer types, and PDAC is hypocellular compared to many other tumors, with a median cellularity of 26% [102,103]. Biopsies obtained via EUS-FNA are small and do not always supply sufficient material for sequencing [104]; by their nature, these cytology samples disrupt tissue architecture and would not permit TME assessment. A few examples of pancreatic core biopsies are shown in Figure 3. Although tissue architecture is preserved, these biopsies often show only a small fraction of tumor cells, with a predominance of blood and fragments of nonneoplastic pancreatic parenchyma in the background. Contaminating gastrointestinal mucosa can be seen. There is often tumor-associated desmoplastic stroma, but this can be challenging to differentiate from the fibrosis and atrophy seen in the background pancreas, which are common concurrent findings in PDAC patients. The hypocellularity, small size, artifacts, and mimics may complicate assessment of the TME from biopsy material. 

Another question is whether small biopsy samples are representative of the entire tumor given regional and microenvironmental heterogeneity. In one study, Ino et al. compared the level of tumor-infiltrating lymphocytes in PDAC resections to simulated FNA “biopsies” taken from the resection specimen, finding that the level of CD8+ T cells was only moderately correlated (r = 0.46) [100]; the group estimated that averaging measurements from five biopsies would be required to achieve a near-perfect correlation. Further highlighting possible sampling issues, Tahkola et al. found that measurements of immune infiltration in tumor microarray-like “hotspots” were more variable and overall inferior to whole-slide measurements for prognostication [101]. 

Furthermore, pathologists frequently need to allocate precious biopsy material for a variety of purposes, including diagnosis, ancillary tests for mismatch repair deficiency, and tumor molecular profiling for potentially actionable somatic findings. Tissue may be limited for additional studies that may be needed to characterize the TME. 

#### 5.1.2. TME Differences between Primary and Metastatic Sites

Few studies have compared TME characteristics between primary and metastatic PDAC lesions. One study by Whatcott et al. found similarities among certain characteristics; in a small cohort of seven paired primary and metastatic tumors, the degree of desmoplasia was similar [98]. A rudimentary analysis of a larger cohort of over one-hundred non-paired metastatic and primary lesions suggested that the total percent areas of various collagens and SMA+ stromal cells were statistically indistinguishable or only marginally different [98]. In contrast, another study reported that metastatic lesions had a lower tumor stromal density than paired primary PDAC lesions [97]. Comparison of metastatic (liver) and pancreatic TME using single-cell RNA sequencing showed that mesenchymal cells differed by site, while immune populations had a similar distribution [7]. There also appear to be differences among metastatic sites, with PDAC lung metastases showing greater immune cell infiltration than liver metastases in a PDAC mouse model and small cohort of paired human samples [99]. Overall, it remains an open question whether histologic assessment of the TME in a metastasis can provide information that is applicable to the primary site as well. 

#### 5.1.3. Gross and Microscopic Evaluation of PDAC Resections

Surgical resections are the other main category of specimens pathologists evaluate in PDAC patients. Resection specimens require expert gross evaluation to ensure proper orientation and margin assessment, determine the size and extent of the tumor, and provide proper sections for microscopic analysis. Current grossing protocols for resections are cited in the College of American Pathologists (CAP) protocol [105,106]. PDAC masses often show indistinct borders on gross examination and may show gross features similar to fibrosis secondary to chronic pancreatitis or pancreatic atrophy. Because of this, sections to assess margins and determe the extent of the tumor are carefully and thoroughly submitted for microscopic examination. As the importance of TME assessment increases, considerations may include the diagnostic difficulty in differentiating TME changes from background pancreatitis and the significance of TME changes at margins. Will current grossing protocols need to change to accommodate future TME-directed therapies? 

Microscopic evaluation protocols are specified by governing bodies such as the CAP. Currently, microscopic evaluation is used to confirm tumor size and stage, evaluate histologic diagnosis of PDAC with histologic type and grade, and report margins and lymph node status. No further subtyping is currently standard. Pathologists also report the presence or absence of PNI and LVI (Figure 3). Reporting the presence and extent of tumor-infiltrating lymphocytes is not standard, as is the case in melanoma profiles, for example [107]. This reporting could change dramatically with the emergence of TME-directed therapies, as we discuss in more detail below.

#### 5.1.4. Effect of Neoadjuvant Treatment on the PDAC TME

Another consideration as pathologists evaluate tissue specimens is the potential impact of neoadjuvant treatment on tumor cells and the TME. Neoadjuvant regimens are becoming more frequent at high-volume centers, potentially affecting the assessment of resection specimens or biopsies for metastatic disease or local disease recurrence. Typically, neoadjuvant therapy is recommended for patients with borderline resectable and locally advanced disease with no distant metastasis with a goal of downstaging and achieving more definite surgical resection. Neoadjuvant therapy may also be recommended in certain localized resectable tumors to reduce the risk of intraoperative tumor spillage and sterilize the lymph nodes and may improve response to postoperative treatment. However, there is limited evidence to recommend specific neoadjuvant regimens off-study, and practices vary regarding the use of chemotherapy and radiation [108].

In cases of neoadjuvant therapy prior to resection, pathologists evaluate the treatment effect percentage. CAP recommends reporting of the generic modified Ryan scheme tumor regression score [109], although in PDAC a three-tier system (no residual, less than 5% residual tumor, or >5% residual tumor) was found to correlate with disease-free survival [10,11].

Neoadjuvant treatment otherwise affects the tumor cells and TME in several ways. Chemotherapy affects residual tumor cell gene expression [110]. Neoadjuvant chemotherapy was associated with a decrease in intercellular interactions (as predicted by the computational inference of ligand–receptor interactions from single-cell RNA sequencing data), including a reduction in the inhibitory checkpoint molecular expression of CD8+ T cells and cancer cells (TIGIT-PVR signaling), indicating that chemotherapy alters the TME in PDAC [92]. These results imply that chemotherapy changes TME behavior and may alter responsiveness to TME-directed therapies. In the same study, there was variation in the proportion of myofibroblastic and inflammatory CAFs across 27 patient tumors, with no correlation to the proportion of classical vs. basal-like cancer cells (i.e., independent of Moffit PDAC subtype) [92]. Additionally, some have suggested that neoadjuvant chemotherapy may affect stromal ratio, depending on therapy type [111]. In routine practice, several pathologic changes are noted more frequently in tumors that are analyzed following neoadjuvant therapy. Pathologic features reportedly associated with therapeutic effects include foamy gland changes, mucus lake (mucin pool) formation, fibrosis, foamy macrophages, cholesterol clefts, and calcifications (Figure 4B–D) [112,113]. How these substantial treatment-induced microenvironmental changes could affect the response to TME-directed therapies remains an open question.

### 5.2. Heterogeneity of the PDAC TME

#### 5.2.1. Inter-Patient Heterogeneity

Molecular studies demonstrate that, like the mutational landscape of tumor cells, the TMEs of different PDAC tumors are heterogeneous. In one study, Moffitt and colleagues applied a “virtual microdissection” approach to bulk microarray data from PDAC tumors and normal samples, computationally identifying two stromal subtypes that are independently prognostic, termed “normal” and “activated” [6]. “Activated” stromal gene signatures corresponded to a 38% reduction in median survival compared to the “normal” subtype [6]. The “normal” stroma signature is characterized by pancreatic stellate cell (PSC) markers and expression of smooth muscle actin, vimentin, and desmin; a diverse group of genes driving an “activated” stromal signature included fibroblast activation protein (FAP), the integrin ITGAM, chemokines CCL13 and CCL18, macrophage-associated genes, WNT family members WNT2 and WNT5A, MMP9, MMP11, and SPARC [6]. In PDAC xenograft models, mouse stromal cells recapitulated elements of “activated” and “normal” signatures; however, engrafted tumors exhibit a bias toward the “activated” signature, and it remains unclear how faithfully the murine stromal cells recapitulate the human TME [6]. Interestingly, there was no statistical association between tumor subtype and stromal subtype—rather, each stromal subtype was seen to occur in “classical” and “basal-like” tumor subtypes and was independently linked to prognosis [6].

In a separate transcriptomic study of formalin-fixed paraffin-embedded (FFPE) PDAC specimens, Puleo et al. identified four independent stromal gene components (activated stromal, structural vascular, inflammatory stromal, and inflammatory) [114]. While tumor-cell-specific gene signatures validated the previously described “classical” and “basal-like” subtypes, they described two additional subtypes that arose from low-tumor-burden samples with high stromal content: “stroma-activated” PDAC was characterized by high expression of *ASMA*, *SPARC*, and *FAP* [114], whereas “desmoplastic” PDAC was characterized by high expression of structural and vascularized stroma components [114]. This study corroborates Moffitt et al. in demonstrating that the PDAC TME differs between patients. Interestingly, they identified an opposite prognostic impact of the TME depending on whether the tumor had a classical or basal-like subtype; non-immune stroma subtypes had negative prognostic impact on classical tumors, but a positive prognostic impact on basal-like tumors [114]. Collisson et. al. proposed a harmonized nomenclature including two stromal subtypes (activated and normal) that accompany the different tumor-cell subtypes [86]. Overall, these molecular studies established that important TME differences exist between patients and demonstrate that prognostically relevant information can be gleaned from TME transcripts that “contaminate” a bulk transcriptomic study.

#### 5.2.2. Intra-Tumoral Heterogeneity

The TME is not only heterogeneous between patients, but also within a single PDAC tumor. Grünwald et al. reported that spatially confined “sub-TMEs” exist in PDAC, and that intratumoral variation in TME was a poor prognostic indicator [18]. Neoadjuvant treatment affected the TME, which was less likely to show a “reactive” state and more likely to show a “deserted” or intermediate state [18]. There are practical challenges to evaluation of sub-TMEs; evaluation of sub-TMEs in biopsies may be impossible due to sampling (the authors did not attempt to quantify sub-TMEs in biopsies, only resections). Furthermore, although sub-TMEs were initially described by H&E morphology, the exact criteria were not precisely defined. 

### 5.3. Nerve 

#### 5.3.1. Perineural Invasion

PDAC is notorious for PNI, and PNI is reported in about 70–80% of cases [115], with some studies reporting an incidence of 100% [13,115]. In comparison to other gastrointestinal malignancies, PNI in PDAC is more prevalent [13]. Although not a factor in TNM staging criteria, perineural invasion is an independent indicator of poor prognosis and is a mandated reporting element in CAP guidelines [106].

A systematic review and meta-analysis found that PNI reduced overall survival (hazard ratio 1.68) and disease-free survival (hazard ratio 2.53) [115]. The prognostic impact of PNI was found to be independent of LVI in PDAC patients who underwent pancreaticoduodenectomy [30]. In an analysis that also included patients who underwent pancreaticoduodenectomy for other malignancies, the adverse prognostic impact of PNI was independent of margin status, tumor size, tumor differentiation, and regional lymph node status [30].

The close association of PDAC with nerves complicates resection and leads to clinical complications such as rapid gastrointestinal transit and pain [116]. PDAC frequently invades posteriorly along autonomic nerves associated with the superior mesenteric artery and celiac axis, sometimes making complete excision impossible. Resection of these autonomic nerves denervates the small bowel and can lead to rapid gastrointestinal transit, nutritional depletion, and dependence on total parenteral nutrition [116]. PNI enables the local outgrowth of tumor cells, and in some cases may provide a conduit to the lymphatic system [117]. The severity of PNI has also been inversely correlated to CD8+ T cell infiltration in human PDAC samples [118].

How should pathologists report PNI, and what exactly constitutes PNI? An early and widely adopted definition of PNI described the phenomenon broadly as tumor cell invasion in, around, and through the nerves [119]. A prominent review by Liebig et al. also advocated for a broad definition of PNI: a tumor in close proximity to nerves and involving at least 33% of its circumference or tumor cells within any of the three layers of the nerve sheath [120]. In our view, pathologists should use the definition(s) of PNI that were applied in the studies that determined its clinical significance in each malignancy. In PDAC studies, PNI was defined more narrowly as viable tumor cells within the perineural space [12,14,30]. The presence of acellular mucin within a nerve was not sufficient to call PNI, nor was the presence of neoplastic glands abutting a nerve [12]. 

A handful of studies found that more prognostic information may be gleaned by moving beyond dichotomous assessment of PNI (present vs. absent). For example, the size of involved nerves correlates with poorer margin status; PDAC cases with PNI involving larger nerves > 0.8 mm were approximately fourfold more likely to have positive resection margins compared to cases where PNI involved only nerves ≤ 0.8 mm [12]. Intraneural invasion, defined as tumor invasion into the axon of the nerve, was associated with 1.3-fold increased frequency of recurrence compared to cases with PNI but not intraneural invasion [12]. Whether the PNI was intratumoral or extratumoral, or intrapancreatic vs extrapancreatic, did not seem to affect clinical outcomes [12]. The location of tumors within the pancreas (head vs. body-tail) also did not correlate with differences in PNI in one preliminary analysis [14].

The extent of PNI may also hold additional prognostic significance: Liebl et al. formulated a neural invasion “severity” score encompassing epineural, perineural, and intraneural involvement [13]. By this metric, tumors with more severe neural invasion correlated with worse overall survival. Assessment of a similar PNI scoring system found that while PNI severity is associated with more aggressive disease, PNI severity was not significant in multivariate analysis [14].

For now, the simple presence or absence of PNI remains the standard. Methods evaluating more detailed assessment of PNI will require prospective validation, which is anticipated in the ongoing VANISSh trial (NCT04024358). Future studies should clearly state their definition of PNI. While sophisticated scoring algorithms would be more burdensome to pathologists, studies have demonstrated that artificial intelligence (AI) can aid in the detection of PNI [121], including one study where AI-assistance augmented PNI detection by pathologists from 52% to 81% of PDAC cases [122].

#### 5.3.2. Nerve–PDAC Interactions

The topic of nerve–PDAC interactions has been reviewed extensively [123,124,125]. Both PDAC and chronic pancreatitis (a known PDAC risk factor) are associated with nerve hypertrophy and increased neural density [126]. The pancreas is innervated by a variety of nerve fiber types which have distinct interactions with PDAC.

Sensory and parasympathetic nerves are generally associated with a pro-tumor role. Notably, ablation of sensory neurons during the neonatal period delays development of precursor PanIN lesions and improves survival in a mouse model of PDAC [127]. Sympathetic activity, including chronic stress, promotes PDAC growth; experiments in animal models have linked this to direct adrenergic-mediated proliferation of PDAC cells [128], as well as immunosuppressive effects [129,130]. Sensory and sympathetic axons within the PDAC microenvironment metabolically support PDAC by releasing serine, which supports translation in nutrient-depleted cancer cells [131]. In turn, tumor cells secrete NGF, promoting tumor innervation, with inhibition of the Trk-NGF axis reducing the growth of PDAC tumors in mice. Epidemiological studies have uncovered modest yet significant improvements in the survival of PDAC patients who take beta blockers [132,133], who have reduced nerve density on resection [128].

There are conflicting reports on the effect of parasympathetic pathways. In support of a pro-tumor role, acetylcholine signaling was shown to impair the recruitment of CD8+ T cells in PDAC mouse models, with vagotomy improving survival [118]. The density of VACht-labeled parasympathetic fibers was also found to be an adverse prognostic indicator that was associated with tumor budding in one study [15]. In contrast, other studies found that vagotomy accelerated PDAC in mouse models [134,135].

Schwann cells may also have a role in the PDAC TME. Co-culture with Schwann cells induces PDAC cell migration and invasive behavior in vitro [136,137]. The area of Schwann cells in histologic sections was found to be an adverse prognostic indicator in one PDAC cohort [16]. Spatial transcriptomics analysis suggested that perineural areas had an increased basal-like tumor cell signature and a more inflammatory cancer-associated fibroblast (iCAF) signature; areas further from nerves harbored classical- and myofibroblast-related signatures [16], suggesting that Schwann cells induce the conversion of tumor cells and cancer-associated fibroblasts (CAFs) to alternative states.

In summary, the interactions between PDAC cells and various nerve types are intricate and potentially targetable. Identification of PNI within resection specimens is the current standard given its prognostic significance, and studies should clearly define and ideally harmonize their criteria for PNI. Additional biomarkers including PNI severity, nerve hypertrophy, parasympathetic innervation, and Schwann cell area have been described; however, the limited additional prognostic value and added complexity may limit their adoption. 

### 5.4. Fibroblasts

#### 5.4.1. Cancer-Associated Fibroblast Populations

Fibroblasts are the predominant cell type within the PDAC TME, and thus considerable attention has been dedicated to understanding stromal fibroblast populations. Fibroblasts are largely responsible for the deposition of extracellular matrix within PDAC; however, diverse subtypes of these cancer-associated fibroblasts (CAFs) are being investigated with possible inflammatory and/or anti-inflammatory properties. The earliest subtypes of CAFs identified were a myofibroblast-like population (myCAF) and an inflammatory fibroblast population (iCAF) [138]. Additional populations of CAFs include fibroblasts with antigen presentation properties similar to myeloid populations, termed antigen-presenting CAFs [139]. 

CAFs have also been subtyped by the expression of certain cell surface markers. Single-cell mass cytometric analysis has identified the marker CD105 to differentiate functionally distinct populations of CAFs, including tumor-permissive CD105+ CAFs and tumor-suppressive CD105− CAFs [19]. These fibroblast compositions could potentially be delineated by pathologists utilizing IHC stains for the cell surface marker CD105. However, tumor heterogeneity could make this assessment difficult, and the TME may vary among PDAC patients depending on the influence of genetic mutations. In addition, it may be difficult to ensure that CD105 is expressed on a fibroblast without multiplex IHC, which poses technical, logistical, financial, and regulatory challenges.

Another CAF population with functional relevance is those expressing fibroblast activation protein (FAP) and high levels of CXCL12, which inhibits T cell infiltration. Depletion of this subtype of CAFs sensitizes PDAC tumors to immune checkpoint therapy in pre-clinical models [20]. A subset of CAFs in PDAC mouse models also express LRRC15, which was not expressed in normal pancreas and was postulated to promote tumor growth and inhibit anti-tumor immune responses [140].

While the exact functions of various CAF subgroups remain under investigation, there is optimism that targeting certain CAF populations may potentiate existing systemic therapy, either via modulating the extracellular matrix or promoting the immune response. For example, hedgehog signaling contributes to desmoplasia in PDAC, and inhibition of hedgehog signaling improves the delivery of chemotherapeutics to PDAC tumors [141]. Pathologists may be called upon to perform ancillary studies such as IHC to characterize the CAF populations within a tumor for therapeutic purposes. 

#### 5.4.2. Focal Adhesion Kinase

One target that has emerged in the fibrotic PDAC TME is Focal Adhesion Kinase 1 (FAK). It has been reported that around 80% of PDAC tumors express FAK and phosphorylated FAK, both in stromal cells and particularly in tumor cells [9]. A series of preclinical experiments supported a model whereby FAK activity in PDAC cancer cells led to cytokine production that promoted a fibrotic and immunosuppressive TME rich in CAFs, myeloid-derived suppressor cells, and regulatory T cells (Tregs); small-molecule inhibition of FAK reversed these phenomena and led to an increase in CD8+ T cells that controlled tumor growth, as well as a potentiated response to immune checkpoint inhibition. A subsequent study by the same group provided evidence that the fibrotic PDAC stroma limits the efficacy of radiotherapy in mouse models by reducing interferon signaling, apoptosis, and immune priming [142]; FAK treatment “reprogrammed” CAFs, rescued radiation-induced interferon signaling, and promoted anti-tumor immune responses. Whether FAK inhibition can augment responses to radiotherapy and checkpoint inhibition in PDAC is currently under investigation in clinical trials (NCT04331041, NCT03727880). FAK has also been found to have a kinase-independent function in antigen presentation on MHC-1 that had a stronger association with the “classical” PDAC subtype [143]. Generally, high expression of FAK is associated with worse overall survival across solid carcinomas [144], although studies evaluating FAK expression in PDAC have not shown an association with survival [145,146]; further studies evaluating whether FAK expression has prognostic or treatment implications in PDAC may be warranted given the emergence of FAK inhibition as a promising therapeutic strategy.

In addition to the potential biomarkers listed above, characterizing the unique CAFs of PDAC may help aid pathologists diagnostically. As mentioned previously, PDAC tumor detection can be difficult in small pancreatic biopsies. An immunostain that could distinguish CAFs from the reactive fibroblasts found in chronic pancreatitis could facilitate detection in otherwise non-diagnostic biopsies.

#### 5.4.3. Tumor–Stroma Ratio

Another stroma-related biomarker with potential clinical significance is the evaluation of tumor–stroma ratio. Tumors with a low tumor–stroma ratio (i.e., tumors that have more tumor cells than stroma) had worse prognosis in one study analyzing whole-slide images [17]; however, another study that quantified a 10× field near the invasive front found that tumor–stroma ratio had no association with prognosis [147].

### 5.5. Metabolism and Autophagy 

#### 5.5.1. Altered Metabolism in PDAC

PDAC metabolism is an extensive topic that has been reviewed elsewhere [148]. The fibrotic, hypovascular PDAC TME is associated with hypoxia and altered nutrient profiles, including a relative scarcity of glucose, glutamine, and serine and abundant amino acids compared to neighboring benign pancreatic parenchyma [149]. Oncogenic *KRAS* mutations lead to substantial metabolic reprogramming in cancer cells [150], and PDAC exploits macropinocytosis [151], extracellular vesicle signaling [152], and autophagy [153] to scavenge nutrients and flourish. Metabolic “subtypes” of PDAC have been described, and metabolic heterogeneity has been described at the inter- and intratumoral level [154,155].

#### 5.5.2. Metabolic Interplay of the PDAC TME

The TME engages in complex metabolic crosstalk with cancer cells that includes competing for nutrients, sharing nutrients, and utilizing metabolites as signaling molecules (reviewed in [156]). There are numerous ways in which TME metabolites support tumor growth. Alanine secreted from pancreatic stellate cells (PSCs) is a major source of carbon for PDAC cells, decreasing their dependence on glucose and nutrients from the blood supply [157]. PDAC also scavenges nutrients from extracellular matrix components, such as collagen and hyaluronic acid [151,158,159], as well as from exosomes [152,160]. Further, PDAC-derived TGF-β reprograms CAFs to catabolize branched-chain amino acids and secrete branched chain α-ketoacids that support PDAC growth [161].

Metabolites also serve as signaling molecules in the PDAC TME. PSCs secrete phosphatidylcholines which are converted into the potent signaling molecule lysophosphatidic acid in the extracellular space, fueling PDAC growth [162]. Conversely, the KRAS-dependent production of lactate and G-CSF were found to be crucial for the programming of TAMs [163]. 

#### 5.5.3. Nutrient Recycling and Autophagy

PDAC is known to have autophagy addiction or an increased engagement of the “self-eating” recycling program [164,165,166]. Besides its function in organelle maintenance, autophagy also plays a crucial role in regulating nutrient availability within PDAC cells. This reliance on autophagy leads to selective vulnerabilities in PDAC cells. Mukhopadhyay et al. found that the oxidized cysteine (cystine) transporter SLC7A11 requires autophagy machinery for proper membrane localization, and inhibition of autophagy restricts cystine availability and slows the growth of PDAC cells [167]. Mechanistically, depletion of SLC7A11 selectively leads to PDAC ferroptosis [168], a means of programmed cell death whereby iron catalyzes catastrophic lipid peroxidation. 

This example is but one of many complex roles for autophagy that have been described in PDAC initiation and development (reviewed in [169]). Clinical trials of autophagy inhibitors in PDAC have yet to show efficacy; one phase II trial found that hydroxychloroquine monotherapy reduced autophagy but had negligible efficacy in patients with previously treated metastatic PDAC [170].

It was discovered that the TME promotes resistance to autophagy inhibition in PDAC by buffering the nutrient supply. For example, PDAC requires autophagy of ferritin (ferritinophagy) for iron homeostasis; yet, during autophagy inhibition, CAFs compensate for the drop in labile iron [171]. Furthermore, genetic abrogation of ferritinophagy led to increased CD68+ macrophage infiltration in a PDAC mouse model, suggestive of further nonautonomous compensatory mechanisms [172]. The specific transporters required for this nutrient sharing may be targetable [173], suggesting that combinatorial approaches could overcome PDAC resistance to autophagy inhibition. In another example of TME metabolites thwarting PDAC treatment, the release of pyrimidines from PSCs and TAMs was found to promote gemcitabine resistance [174,175].

A handful of prognostic markers related to metabolism and autophagy have been identified. Expression of GFAT1, the rate-limiting enzyme of hexosamine biosynthesis, correlates with poor prognosis [176]. An elevated ferritinophagy gene signature and elevated protein levels of Ferritin Heavy Chain 1 (FTH1) were each associated with poor prognosis in PDAC [172]. Another report found that poor prognosis was associated with the expression of the transcription factor EB (TFEB), a central regulator of autophagy, as well as its direct target RAB5A [177].

### 5.6. Immune Cells

PDAC is an immunologically “cold” tumor. Myeloid cell populations are the largest immune constituent in PDAC tumors. This includes neutrophils, which are thought to support an immune-suppressive tumor environment [178,179,180,181,182]. Promising pre-clinical data have emerged via targeting CD11b/CD18 reception on tumor-infiltrating myeloid subsets [24]. Another approach is to utilize CD40 agonists to program tumor-associated macrophages to a less immunosuppressive status, ultimately leading to CD8 T cell infiltration with possible implications for more successful immunotherapy options [26,27,183]. The immune-suppressive tumor environment in PDAC additionally shows a lack of antigen-presenting dendritic cells, and increasing this population of immune cells is also a therapeutic interest [25,184]. For all of the above, pathologists may be asked to identify these components of the immune system via IHC.

Several studies have found that adaptive immune cells have prognostic significance in PDAC. In a study that employed multiplex IHC imaging with digital image analysis to examine the spatial relationships of PDAC cells and immune cells, infiltration of PDAC by T cells (total T cells, CD8+ cytotoxic T cells, and CD4+ T effector cells) was found to be an independent positive prognostic indicator [21]. Specifically, close spatial proximity of CD8+ T cells to tumor cells (within a 20-micron radius) correlated with improved survival. In another report, CD3+ lymphocytes and high collagen correlated with improved survival in multivariate analysis [185]. A study of long-term survivors of PDAC found patients with both high CD8+ T cell infiltration and high neoantigen quantity to have particularly good prognosis [22]. Neoantigens with predicted cross-reactivity to microbial epitopes appeared to be particularly potent in evoking anti-tumor immunity. Additional evidence for the quantification of immune cells includes studies showing that tumor-infiltrating lymphocytes have a prognostic impact [100,101], including a study showing that the level of CD8+ T cells in PDAC resections was prognostic in a multivariate analysis [23]. Additionally, within PDAC resections, the presence of B-cell-rich tertiary lymphoid tissues, but not isolated B cells, was associated with improved prognosis [29].

Although several studies have found that tumor-infiltrating lymphocytes have a significant prognostic impact in PDAC, it is not standard practice in the United States to comment on lymphocyte abundance. This may be due to the relative recency of these studies, differences in methodology (e.g., lymphocyte type and location), and the limited value of prognostic information in a uniformly aggressive malignancy. In contrast, the lymphocyte reporting in melanoma recommended by CAP is simple (three-tiered qualitative scoring), can be performed on routine histologic sections, and has a large prognostic impact (37% 8-year survival for absent lymphocytes, and 77% survival for brisk infiltrate) [107].

Like other components of the TME, immune cell populations show tumor heterogeneity. In an integrated multi-omic study of PDAC tumors and paired peripheral blood, Steele et al. found that the immune landscapes in each individual patient were heterogeneous, including variable quantities of immune cells [186]. Individual patient tumors showed an inverse relationship between myeloid and CD8+ T cells, consistent with evidence that myeloid cells are immunosuppressive in PDAC. Single-cell RNA sequencing revealed complex, highly variable patient-specific landscapes of immune checkpoint ligand and receptor expression amongst different immune cell types and showed that the complement of immune checkpoint genes in CD8+ T cells was unique [186]. The immune checkpoint receptor *TIGIT* expression was defining of an exhausted CD8+ T cell phenotype in PDAC and, as a key mode of immune suppression of the CD155/*TIGIT* axis, it is proposed as a target to enhance immunotherapy [186,187]. *TIGIT* expression, like other immune checkpoint receptors, was highly heterogeneous across patients. As discussed within other sections of this review, this variability enforces the need for personalized assessment of immune cell populations in PDAC.

### 5.7. Vasculature

Aberrant vascular networks are a hallmark of the cancer TME. Notably, PDAC is hypovascular in comparison to normal pancreatic tissue as well as other gastrointestinal and hepatobiliary tumors [188,189]. The density of tumor vessels in PDAC is lower than in normal pancreas, with a larger caliber and reduced branching. PDAC vessels also harbor unusual basal projections termed basal microvilli, whose significance remains poorly understood [189]. 

Vascular networks have a complex influence on PDAC growth. Of course, LVI is an established indicator of poor prognosis in PDAC and a mandated component of the CAP synoptic reporting template [30,31]. Although some studies found no correlation between PDAC vascular morphology and patient outcome [188,189], others have suggested an association: one study examining the TCGA dataset found that high CD31 expression correlated with improved overall survival in PDAC [190], and a spatial analysis of microvascular density in and around PDAC tumors found certain morphologic correlated to prognosis [32]. Some analyses of human PDAC samples found that poorly differentiated tumors had greater vascular density and less stroma compared to differentiated tumors [188,191], though other studies found no relationship [189,192]. One possible explanation for the discrepant findings may be the subjectivity involved in selecting and measuring the vasculature. Automated computational analysis methods may clarify whether vascular parameters correlate with other pathologic factors and patient outcomes in PDAC. These conflicting reports may reflect the contrasting effects of the hypovascular TME on tumor growth, on one hand contributing to tumor hypoxia and poor penetration of drugs [141], but also starving the growth of cancer cells by limiting their uptake of nutrients. 

Efforts to target the PDAC vasculature have yet to be successful in the clinic but remain under investigation. Anti-angiogenic therapy with the VEGF-targeting antibody bevacizumab has not shown clinical benefit, with or without concomitant chemotherapy [193,194]. Recent clinical trials found that multi-tyrosine kinase inhibitors have been efficacious in treating a variety of other neoplasms [195], including pancreatic neuroendocrine tumors [196], which, in contrast to PDAC, are notably hypervascular [197]. One such drug, surufatinib, impairs angiogenesis by targeting VEGFR while simultaneously blocking FGFR1 (a mechanism of resistance to anti-VEGF therapy) and CSF-1R, thereby decreasing the recruitment of tumor-associated macrophages [198]. Surufatinib is currently being evaluated in a clinical trial for PDAC (NCT05832892). 

The markedly desmoplastic stroma of PDAC appears to prevent the formation of a robust vascular network. Efforts to reverse desmoplasia in PDAC thus uncovered a complex influence of tumor vasculature. Genetic deletion of *Sonic Hedgehog Signaling Molecule* (*Shh*) or pharmacologic inhibition of SHH in various PDAC mouse models abrogated the desmoplastic stroma and reduced the numbers of ɑSMA+ myofibroblasts and myeloid cells in the TME, but those treatments concurrently led to an increase in tumor vascularity and actually accelerated tumor growth [191,199]. These SHH-deficient mouse tumors were sensitive to anti-angiogenic therapy targeting VEGF signaling. Such studies highlight an inverse relationship between the desmoplastic stroma and vascular networks and show that a more robust tumor vasculature promotes tumor growth while also potentially enhancing the delivery of therapeutics [141]. The interrelatedness of TME attributes and their conflicting pro-tumor and anti-tumor roles illustrate the challenge of targeting TME components and explain why clinical trials examining hedgehog inhibition in PDAC have thus far been unsuccessful [200,201].

Although the prognostic significance of vascular structure in PDAC is unclear, one consistent finding is the variation in the degree of angiogenesis, both in mouse models as well as in patients. KIC mice (oncogenic *Kras*, p16 loss), KRC mice (oncogenic *Kras*, *Rb1* loss), and KTC mice (oncogenic *Kras*, *Tgfbr2* loss) exhibit angiogenesis [192,202,203], whereas KPC mice (oncogenic *Kras*, p53 mutant) and KPfl+C mice (oncogenic *Kras*, p53 loss) develop hypovascular tumors [141]. In patient samples, Gore et al. reported that 12% (10/85) of PDAC tumors within the TCGA database had increased expression of angiogenesis markers, and analysis of tumor microarrays revealed strong immunoreactivity for endothelial markers CD31 and CD34 in 15/54 tumors [192]. The group also found that endothelial cells supported PDAC growth in the KRC mouse model of PDAC, which was suppressed by the JAK inhibitor ruxolitinib. Thus, variations in PDAC vascular complexity present both a challenge and a possible opportunity; the aforementioned data from preclinical models suggest that certain tumors (perhaps stroma-poor, poorly differentiated, and vascular-rich tumors) may be susceptible to anti-angiogenic therapy.

## 6. Conclusions

There has been notable advancement in comprehending the biology of pancreatic cancer in recent years, and although these gains have not yet led to a transformative impact on clinical treatment for most patients, multidisciplinary approaches show promise. With advancements in the understanding of the diverse genomic landscape and TME, pathologists can expect increasing involvement in this changing field.

As we have emphasized in this review, heterogeneity in PDAC tumors supports the need for a personalized approach to PDAC evaluation and management. Developments in understanding the non-cancerous components of the tumor and their role in driving the aggressive biology and heterogeneity of the disease should be reflected in assessments of an individual patient’s tumor. For the pathologist, this includes providing potentially actionable information within our reports for PDAC patients. This could include characteristics of the TME, such as detailed quantification of features such as the tumor–stroma ratio, the amount of perineural invasion or vascularity of the tumors, and the immune cell populations; TME subtyping via the investigation of specific fibroblast and immune cell populations by marker expression; and newer technologies such as AI and multiplexed IHC to aid in TME evaluation/subtyping. Numerous morphologic and molecular features can be reported that may have prognostic relevance, although many of these prospective biomarkers await prospective validation. Additionally, with newer ancillary studies being requested on biopsy material, pathologists will likely assume a larger role in assessing tissue adequacy and triaging the patient material appropriately. Furthermore, molecular pathologists continue to look upon these advances to include actionable genetic alterations within molecular reports. Overall, pathologists should expect an increased role in providing personalized information on PDAC specimens, with a hope of giving actionable information toward a more manageable disease.

## Figures and Tables

**Figure 1 cancers-16-00794-f001:**
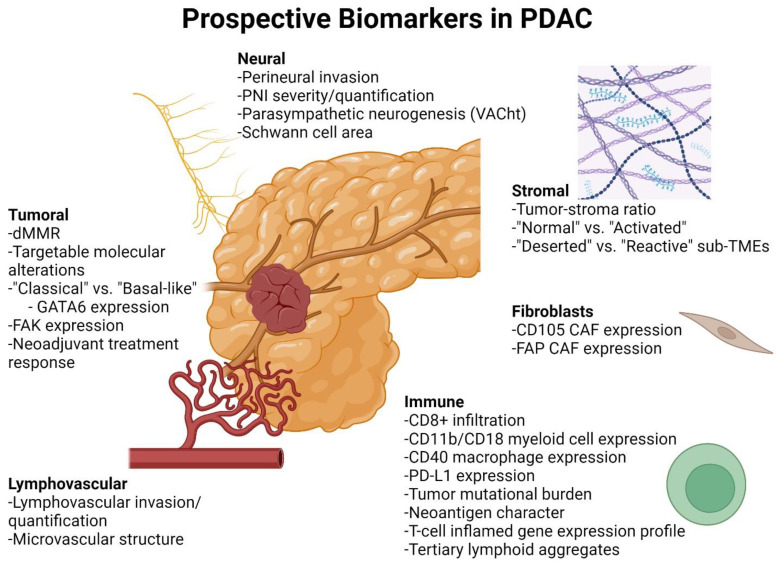
Current and prospective biomarkers for evaluation of PDAC and its TME. PDAC cancer-cell-specific biomarkers include mismatch repair deficiency (dMMR) and targetable molecular alterations [8], “classical” and “basal-like” gene expression profiles [6], FAK expression [9], and neoadjuvant treatment response [10,11]. Nerve-related biomarkers include perineural invasion [12], PNI severity [13,14], parasympathetic neurogenesis [15], and Schwann cell area [16]. Stromal biomarkers include tumor–stroma ratio [17], “normal” vs. “activated” gene expression profile [6], and “deserted” vs. “reactive” sub-TMEs [18]. Fibroblast-related biomarkers include expression of CD105 [19] and FAP [20]. Immune-related biomarkers include CD8+ T cell infiltration [21,22,23], CD11b/CD18 myeloid cell expression [24], CD40 macrophage expression [25,26,27], tumor mutational burden, PD-L1 expression and T-cell-inflamed gene expression profile [28], neoantigen character [22], and tertiary lymphoid aggregates [29]. Vascular biomarkers include lymphovascular invasion [30,31] and microvascular structure [32]. Created with BioRender.com.

**Figure 2 cancers-16-00794-f002:**
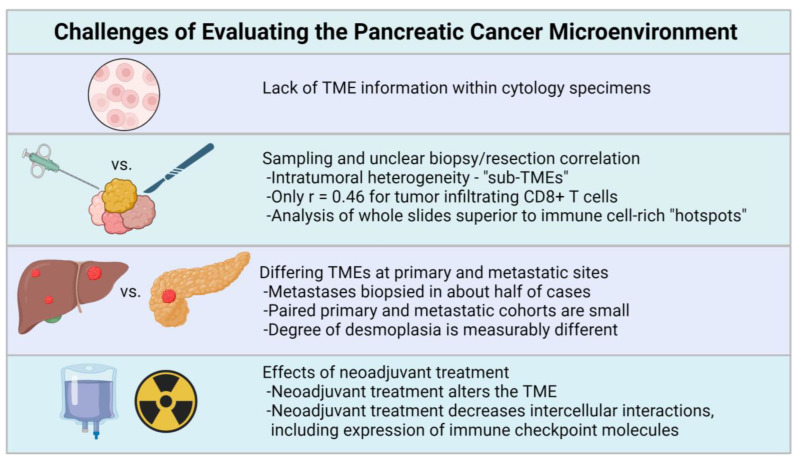
General considerations for pathologic evaluation of the PDAC TME include the lack of TME information in cytology specimens, sampling issues pertaining to small biopsy specimens [18,92], possible TME differences between primary and metastatic sites [39,97,98,99], and the effects of neoadjuvant treatment [18,100,101]. Created with BioRender.com.

**Figure 3 cancers-16-00794-f003:**
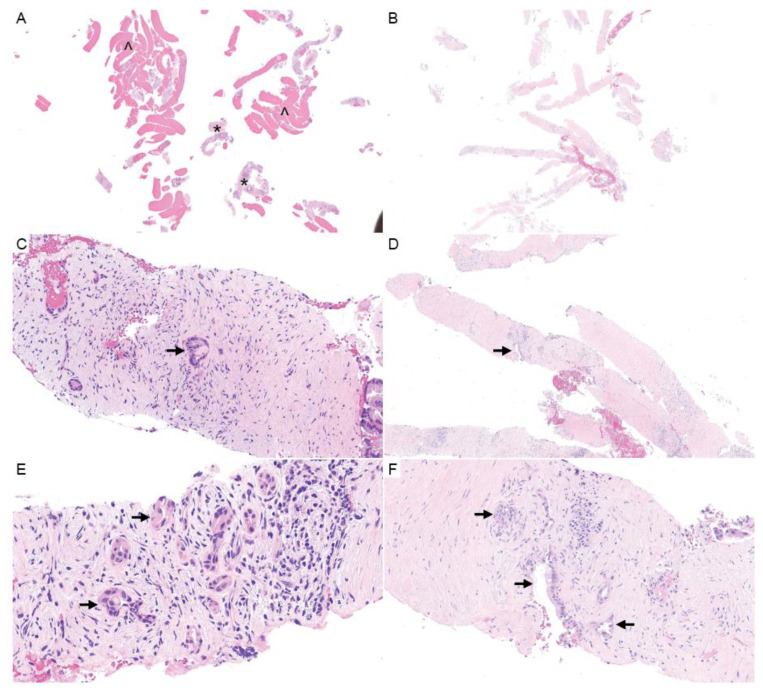
Pancreatic core needle biopsies. (**A**,**B**) H&E (1×) shows the general composition of pancreatic needle biopsies, which may include large amounts of blood (^) with fewer cores of pancreatic tissue (*) (**A**), or predominately hypocellular stroma (**B**). (**C**–**F**) Carcinoma cells may represent a small percentage of the overall biopsy volume (H&E; **C**,**F**, 20×; **D**, 10×; **E**, 40×). The arrows designate tumor cells.

**Figure 4 cancers-16-00794-f004:**
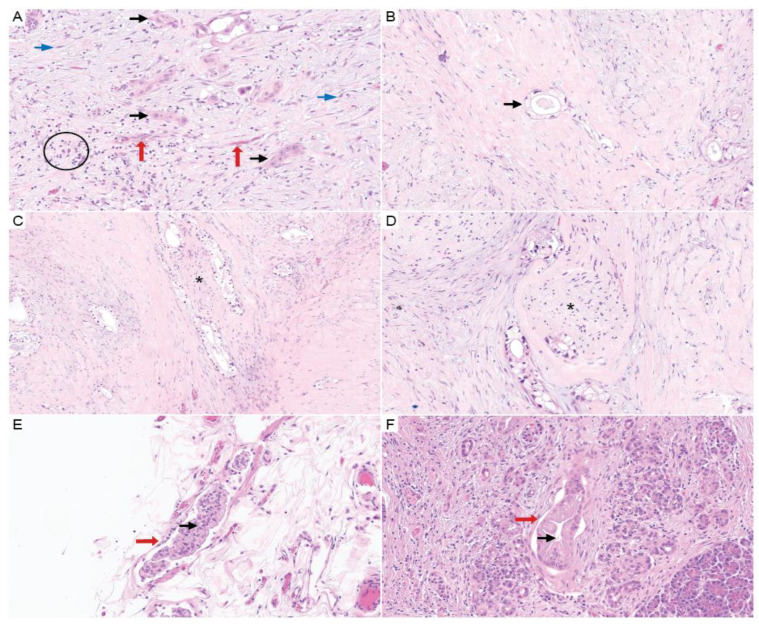
Surgical resection specimen findings in PDAC. (**A**) H&E (20×) shows PDAC (black arrows) with TME constituents of lymphocytic inflammation (black circle), lymphovascular spaces (red arrows), and stromal fibroblasts (blue arrows). (**B**–**D**) Images show foamy gland change, which is associated with neoadjuvant therapy (tumor cells designated by black arrows). PNI is also demonstrated in (**C**,**D**) (asterisk denotes nerves). (H&E; **B**, 20×; **C**, 10×; and **D**, 10×). (**E**,**F**) H&E (20×) shows LVI identified on resection specimens (vessels are marked by red arrows and the black arrows show tumor within the lymphovascular spaces).

**Table 1 cancers-16-00794-t001:** Common genomic drivers and alterations studied in PDAC.

Alteration/Category	Morphologic Correlate	Translational Impact	Pathologist Role	Publication(s)
*CDKN2A*	Altered CDKN2A/p16 associated with adenosquamous or PDAC with complex component	None	p16 IHC interpretation	Schlitter et al., 2017 [84]
*SMAD4, TP53*	Unknown	None	SMAD4, p53 IHC interpretation	Schlitter et al., 2017 [84]
*KRAS* altered	-Classical PDAC -Intestinal type IPMN and colloid carcinomas (GNAS and KRAS)	FDA-approved for small-molecule inhibitors targeting *KRAS*^G12C^, others (including pan-*RAS* inhibitors) in development	-Tissue adequacy -KRAS genotyping	Cox et al., 2014 [50], Moore et al., 2020 [51], Tang and Kang 2022 [52], Hallin et al., 2022 [53], Koltun et al., 2021 [54], Arbour et al., 2023 [93]
*KRAS* wildtype	PDAC variants (colloid, papillary, medullary, tubular) more frequent in KRAS wildtype	10% of PDACs that are KRAS wildtype have distinct molecular profiles and are more likely to have fusions, which are frequently actionable	-Tissue adequacy -KRAS genotyping	Schlitter et al., 2017 [84], Topham et al., 2022 [55]
*BRCA1/2*	Unknown	May benefit from PARP inhibitors	Tissue adequacy	Golan et al., 2019 [61], Momtaz et al., 2021 [62]
SWI/SNF COMPASS-like complex genes (*KDM6A*, *KMT2C*, *KMT2D*, *KMT2A*, *KMT2B*)	Correlate with poor differentiation, squamous features, aggressive behavior, and increased concurrent TP53 mutations	Activin A might be used as a therapeutic target for KDM6A- or KMT2D-deficient PDACs	SMARCA4/A2 IHC interpretation	Andricovich et al., 2018 [68], Yi et al., 2022 [70], Hissong et al., 2023 [71], Lu et al., 2023 [94]
Alternative drivers: *ALK*, *TRK*, *RET*, *NRG1*, *EGFR*	Unknown	-Typically fusions, younger patients -Many are actionable targets	-Tissue adequacy -NGS panel	Kato et al., 2017 [56], Solomon et al., 2020 [57], Okamura et al., 2018 [58]
Mismatch-repair-deficient (dMMR)	Medullary or mucinous/colloid histology	May benefit from immune checkpoint blockade	-MMR IHC interpretation -MSI testing	Goggins et al., 1998 [74], Laghi et al., 2012 [75], Wilentz et al., 2000 [76], Marabelle et al., 2020 [77], Le et al., 2017 [72]
Transcriptional subtypes	Basal like—loss of GATA6 by IHC	Basal like—less responsive to first-line 5-fluorouracil-based therapies	GATA6 IHC interpretation	O’Kane et al., 2020 [88], Duan et al., 2021 [89], Chan-Seng-Yue et al., 2020 [90]
Molecular testing for early detection	N/A	-Circulating biomarkers or cystic analysis -PanIN/IPMN/MCN -Cystic lesions	Tissue/fluid triaging	Singhi et al., 2021 [95], Paniccia et al., 2023 [96]

**Table 2 cancers-16-00794-t002:** Selected clinical trials targeting the PDAC TME.

TME Target	Molecular Agent	Biologic Hypothesis	Biomarker-Related Inclusion Criteria	Phase	NCT Number(s)
Angiogenesis, Immune	Surufatinib (TKI targeting VEGFR, FGFR, CSF-1R) and KN046 bi-specific Ab targeting PD-L1 and CTLA-4	Test whether Tyr kinase inhibition mitigates angiogenesis and immunosuppression	No biomarker-specific inclusion criteria	1/2	NCT05832892
CAF	TGFβ-B-15 peptide vaccine	Test whether TGFβ-B-15 peptide vaccine potentiates immune checkpoint inhibition	No biomarker-specific inclusion criteria	1	NCT05721846
CAF	Vismodegib (SHH inhibitor)	Test whether hedgehog inhibition reduces stromal fibrosis and increases perfusion	No biomarker-specific inclusion criteria	1	NCT01713218
CAF, Immune (TAM, MDSC, Treg)	Itacitinib (JAK inhibitor), INCB050465 (PI3K-delta inhibitor), pembrolizumab (anti-PD-1)	Multiple hypothesized mechanisms: JAK/STAT signaling hypothesized to expand MDSCs and Tregs; PI3K-delta inhibition hypothesized to decrease immunosuppressive TAMs and disrupt tumor–stromal signaling	No biomarker-specific inclusion criteria	1	NCT02646748
CAF, PSC, Immune (CD8+ T cells)	Proglumide (cholecystokinin receptor antagonist)	Examining effects of cholecystokinin receptor blockade on TME	PDAC with adenocarcinoma as dominant histology	2	NCT05827055
Immune	Influenza vaccination	Test whether flu shot potentiates checkpoint inhibitor therapy	No biomarker-specific inclusion criteria	2	NCT05116917
Immune	Lenvatinib (multi-tyrosine kinase inhibitor)	Test whether the immunomodulatory effects of levatinib contribute to anti-tumor activity	No biomarker-specific inclusion criteria	1/2	NCT05327582
Immune	Olaparib (PARP inhibitor) and durvalumab (anti-PD-L1)	Test whether PD-L1 inhibitors synergize with PARP inhibitors in tumors with homologous repair deficiency	DNA damage repair gene mutation present	2	NCT05659914
Immune	Olaptesed pegol (NOX-A12, CXCL12 inhibitor)	Test whether disruption of the CXCL12–CXCR4 axis promotes T cell anti-tumor responses	No biomarker-specific inclusion criteria	1/2	NCT03168139 (completed)
Immune	Paricalcitol (vitamin D agonist)	Test whether paricalcitol potentiates checkpoint inhibition by sensitizing immune cells	No biomarker-specific inclusion criteria	2	NCT03331562 (completed)
Immune	Plerixafor (CXCR4 antagonist)	Test whether CXCR4 inhibition potentiates response to checkpoint blockade	No biomarker-specific inclusion criteria	1	NCT02179970 (completed)
Immune	Plerixafor (CXCR4 antagonist)	Test whether disrupting the CXCL12–CXCR4 signaling axis increases intratumoral T cells	No biomarker-related inclusion criteria	1	NCT03277209
Immune	Sirolimus (mTOR inhibitor)	Test whether mTOR inhibition inhibits tumor cell proliferation and promotes T cell anti-tumor response	No biomarker-specific inclusion criteria	1/2	NCT03662412
Immune	Tocilizumab (anti-IL-6)	Test whether inhibition of IL-6 alleviates tumor-induced immunosuppression	No biomarker-specific inclusion criteria	2	NCT04258150 (terminated)
Immune (dendritic cells, CD8+ T cells)	Rintatolimod (TLR-3 agonist)	Test whether increasing dendritic cell maturation and CD8 T cell cross-priming with a TLR-3 agonist potentiates anti-PD-L1 immune checkpoint blockade	CA 19-9 < 1000 kU/L	1/2	NCT05927142
Immune (in setting of deficient homologous recombination repair)	Niraparib (PARP inhibitor) and dostarlimab (anti-PD-1 mAb)	Assess TME for immune-related changes following PARP inhibition and immune checkpoint blockade	Germline or tumor BRCA1/BRCA2/PALB2 mutation	2	NCT04493060
Immune (M2 TAM)	Pexidartinib (CSF-1R tyrosine kinase inhibitor)	Test whether CSF-1R inhibition depletes immunosuppressive M2 TAMs and potentiates checkpoint inhibition	No biomarker-specific inclusion criteria	1	NCT02777710 (completed)
Immune (MDSC, Treg)	Zolendronic acid	Test whether zolendronic acid and gemcitabine target MDSC and Treg to improve anti-tumor immune response	No biomarker-specific inclusion criteria	1	NCT00892242 (terminated)
Immune (TAM)	GSK3145095 (RIPK1 inhibitor)	Test whether RIPK1 inhibition promotes adaptive immune infiltration	No biomarker-related inclusion criteria	2	NCT03681951 (terminated)
Immune, CAF	Defactinib (FAK inhibitor)	Test whether FAK inhibition reduces tumor fibrosis and potentiates immune checkpoint blockade	No biomarker-related inclusion criteria	1	NCT02546531
Immune, CAF	Defactinib (FAK inhibitor)	Test whether combining standard chemotherapy with FAK inhibition potentiates anti-PD-1 therapy	Elevated CA 19-9 > 200	2	NCT03727880
Immune, CAF	Defactinib (FAK inhibitor)	Test whether FAK inhibition improves responses to stereotactic body radiotherapy in PDAC	No biomarker-specific inclusion criteria	2	NCT04331041,
Immune, Metabolism	TTX-030 (CD39 inhibitor)	Test whether CD39 inhibition increases pro-inflammatory ATP and reduces immunosuppressive adenosine	No biomarker-specific inclusion criteria	1	NCT04306900, NCT03884556 (completed)
Metabolism, Immune	Epacadostat (IDO1 inhibitor)	Test whether IDO1 inhibition potentiates immune checkpoint blockade	No biomarker-related inclusion criteria	1/2	NCT02600949
Metabolism, Immune	Epacadostat (IDO1 inhibitor)	Test whether IDO1 inhibition potentiates immune checkpoint blockade	No biomarker-related inclusion criteria	1/2	NCT03085914 (completed)
Metabolism (adenosine)	PT199 (anti-CD73 mAb)	Test whether anti-CD73 therapy counters adenosine-mediated immunosuppression	No biomarker-specific inclusion criteria (but includes assessment of CD73, PD-L1, and other biomarkers)	1	NCT05431270
Metabolism (adenosine)	SRF617 (CD39 inhibitor)	Test whether CD39 inhibition increases pro-inflammatory ATP and reduces immunosuppressive adenosine	No biomarker-specific inclusion criteria	1	NCT04336098
PSC	Paricalcitol (vitamin D agonist)	Test whether inactivation of vitamin D signaling reduces PSC activation and fibrosis	No biomarker-specific inclusion criteria	N/A	NCT02030860 (completed)
Tumor acidity, hypoxia	L-DOS47 (anti-CEACAM6 conjugated to urease)	Test whether L-DOS47 treatment increases tumor pH	No biomarker-related inclusion criteria	1/2	NCT04203641

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
