# Peer review of "Clinical Evaluation of the Pancreatic Cancer Microenvironment: Opportunities and Challenges"

_cancers, 2024, doi:10.3390/cancers16040794_

Round 1

Reviewer 1 Report

Comments and Suggestions for Authors

The manuscript by Szczepanski et al. reviewed the opportunities and challenges in the clinical evaluation of the PDAC TME. The authors first briefly discussed PDAC management and specimen acquisition and then described their genomic drivers of PDAC and the potential targeted therapies. Most of the latter paragraphs summarized the basic understanding of PDAC TME, including the evaluation of PDAC TME and the different cell compositions within PDAC TME. The paper sufficiently addresses different points of PDAC TME and provides a valuable summary of PDAC TME from a pathological aspect. This review should be accepted for publication in Cancers. I only have a few minor comments to add.

1.     Page 3, “In particular, there has been recent progress targeting KRASG12D with a current Phase 1 clinical trial in progress (NCT05382559)”. Please include the first Pan-KRAS inhibitor (RMC-6236) generated by Revolution Medicines. This drug is already in phase I clinical trial (NCT05379985) and shows promising results.

2.     Page 4, “Such tumors are characterized by microsatellite instability (MSI-high).” Please include the fascinating progress made by Roche and Vividion Therapeutics. The first-in-class WRN inhibitor (RO7589831) is already in phase I clinical trial against dMMR/MSI-high tumors (NCT06004245).

3.     Table 2. Does this table include all or only some clinical trials targeting the PDAC TME? Please modify the title to make this clear.

4.     Figure 2. Could this figure follow the description order in sections 5.1.1-5.1.4?

5.     Page 20, “One such drug, surufatinib, inhibits VEGFR, FGFR, and CSF-1R, blocking angiogenesis while simultaneously targeting mechanisms ….”. Although FGFR and CSF-1R are the targets of surufatinib, they are not tightly linked to angiogenesis. Please rephrase this sentence.

Reviewer 2 Report

Comments and Suggestions for Authors

The manuscript titled “Clinical Evaluation of the Pancreatic Cancer Microenvironment: Opportunities and Challenges” is interesting but there are certain concerns as follows:

1.       The authors need to discuss the metabolic aspect of PDAC TME. Some of the important references are missing. PDAC is known to have autophagy addiction and researchers have shown in PMID: 33531365 how autophagy inhibition can restrict the cysteine pools in PDAC. Previously, it was shown in PMID: 32241947 cysteine depletion led to ferroptotic changes in PDAC. Clinical significance of autophagy in PDAC should also be mentioned in this context.

2.       In regards to the section on CAF, authors need to focus on recent CAF-PDAC crosstalks. Recently, it has been observed how CAF can modulate the PDAC iron pool size as shown in PMID: 37075122. They showed how the CAFs can compensate the drop of labile iron in PDAC after autophagy inhibition. This study highlighted the significance of iron restricted diet during therapy. Further, PMID: 35771492 reported how the loss of ferritinophagy impacted the bioavailable iron in PDAC. Beside iron, another important study PMID: 30837243 have shown the significance of lipid transfer from CAF to PDAC.

3.       In fig 1, some of the terms are ambiguous like the words “burden”, “profile”- in the figure they should provide the biomarker names and schematic of the pathway involved. The PDAC image is too superficial. No need to put references within the image they may be shifted to legends (this holds true for fig1 & 2 as well).

4.       In table 1 instead of Clinical/Actionable consequences authors may prefer to write Translational impact.

5.       In table 2, the provided NCT numbers are not matching with the publication references and this is confusing. The authors need to clarify this.

Round 2

Reviewer 2 Report

Comments and Suggestions for Authors

Authors have clarified my concerns